# Learning elementary structures
# for 3D shape generation and matching

**Theo Deprelle**[1*], **Thibault Groueix**[1], **Matthew Fisher**[2], **Vladimir G. Kim**[2],
**Bryan C. Russell**[2], **Mathieu Aubry**[1]
[1]LIGM (UMR 8049), École des Ponts, UPE, [2]Adobe Research

## Abstract

We propose to represent shapes as the deformation and combination of learnable elementary 3D structures, which are primitives resulting from training over a collection of shapes. We demonstrate that the learned elementary 3D structures lead to clear improvements in 3D shape generation and matching. More precisely, we present two complementary approaches for learning elementary structures: (i) patch deformation learning and (ii) point translation learning. Both approaches can be extended to abstract structures of higher dimensions for improved results. We evaluate our method on two tasks: reconstructing ShapeNet objects and estimating dense correspondences between human scans (FAUST inter challenge). We show 16% improvement over surface deformation approaches for shape reconstruction and outperform FAUST inter and intra challenge state of the art by 2% and 7%, respectively.

## 1 Introduction

Current surface-parametric approaches for generating a surface or aligning two surfaces, such as AtlasNet [11] and 3D-CODED [10], rely on alignment of one or more shape primitives to a target shape. The shape primitives can be a set of patches or a sphere, as in AtlasNet, or a human template shape, as in 3D-CODED. These approaches could easily be extended to other parametric shapes, such as blocks [22], generalized cylinders [4], or modern shape abstractions [16, 26, 28]. While surface-parametric approaches have achieved state-of-the-art results for (single-view) shape reconstruction [11] and 3D shape correspondences [10], they rely on hand-chosen parametric shape primitives tuned for the target shape collection and task. In this paper, we ask – what is the right set of primitives to represent a collection of diverse shapes?

To address this question, we seek to go beyond manually choosing shape primitives and automatically learn what we call "learnable elementary structures" from a shape collection, which can be used for shape reconstruction and matching. The ability to automatically learn elementary structures allows the shape generator to find a better set of primitives for a shape collection and target task. We find that learned elementary structures correspond to recurrent parts among 3D objects. For example, in Figure 1, we show automatically learned elementary structures roughly corresponding to the tail, wing, and reactor of an airplane. Moreover, we find that learning the elementary structures leads to an improvement in shape reconstruction and correspondence accuracy.

We explore two approaches for learning elementary structures – *patch deformation learning* and *point translation learning*. For patch deformation learning, similar to AtlasNet [11], we start from a surface element, such as a 2D square, and deform it into the learned structure using a multi-layer perceptron [23]. This approach has the advantage that the learned elementary structures are continuous surfaces. Its key difference with respect to AtlasNet is that the deformations, and thus the elementary structures, are common to all shapes. For point translation learning, starting from a fixed set of points, we optimize their position to reconstruct the target objects. The drawback of this approach is that it does

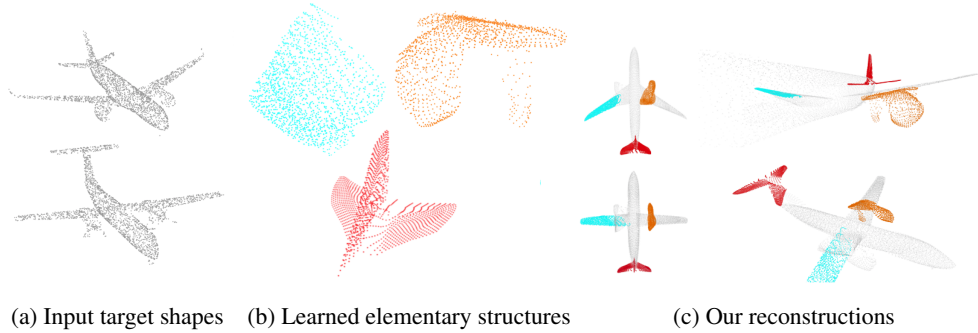

(a) Input target shapes    (b) Learned elementary structures    (c) Our reconstructions

Figure 1: **Problem statement.** We seek to automatically learn a set of primitives (called "learned elementary structures") for shape reconstruction and matching. (a) Input target shapes to reconstruct. (b) Learned elementary structures roughly corresponding to the tail, wing, and reactor of airplanes. (c) Our output reconstructions with learned elementary structures highlighted.

not produce a continuous surface – only a finite set of points. However, this approach is more flexible since it can, for example, change the topology of the structure.

We show how to deform and combine our learnable elementary structures to explain a given 3D shape. At inference, given the learned elementary structures, we learn to position the structures by *adjustment* – a linear (projective) transformation will lead to maximum interpretability, while a complex transformation parameterized by a multi-layer perceptron will make our approaches generalizations of prior shape reconstruction methods [11, 10] using optimized instead of manually defined templates. Moreover, such representation allows for disentanglement of the structure's shape and pose. We include structure learning in a deep architecture that unifies shape abstraction and deep surface deformation approaches.

We demonstrate that our architecture leads to improvements for 3D shape generation and matching – 16% relative improvement over AtlasNet for generic object shape reconstruction and 7% and 2% over 3D-CODED for human shape matching on Faust [5] Intra and Inter challenges, respectively, achieving state of the art for the latter task. Our code is available on our project webpage[1]

## 2   Related Work

Primitive fitting is a classic topic in computer vision [22], with a large number of methods targeting parsimonious shape approximations, such as generalized cylinders[4] and geons [3]. Efficient fitting of these primitives attracted a lot of research efforts [13, 18, 24, 25]. Since these methods analyze shapes independently, they are not expected to use the primitives consistently across different objects, which makes the result unsuitable for discovering a common structure in a collection of shapes, performing consistent segmentation, or correspondence estimation. To address these limitations some methods optimize for consistent primitive fitting over the entire shape collection [15], or aim to discover a consistent set of parts [9, 12, 27]. The resulting optimization problems are usually non-convex, and thus existing solutions tend to be slow, require heuristics, and are prone to being stuck in local optima.

Learning-based techniques offer a promising alternative to hand-crafted heuristics. Zhu *et al.* [31] use a Recurrent Neural Network supervised by a traditional heuristic-based algorithm for cuboid fitting. Tulsiani *et al.* [28] use reconstruction loss to predict parameters of the cuboids that approximate an input shape, and thus do not require any direct supervision. Several recent techniques, concurrent to our work, extend this approach by using more complex primitives that can better approximate the surface, such as anisotropic 3D Gaussians [8], categorie specifique morphable model [14] or superquadrics [20]. All of these techniques use a collection of simple hand-picked parametric primitives. In contrast, we propose to learn a set of deformable primitives that best approximate a collection of shapes.

One can further improve reconstruction by fitting a diverse set of primitives [17] or constructive solid geometry graphs [26]. These methods, however, usually do not produce consistent fitting

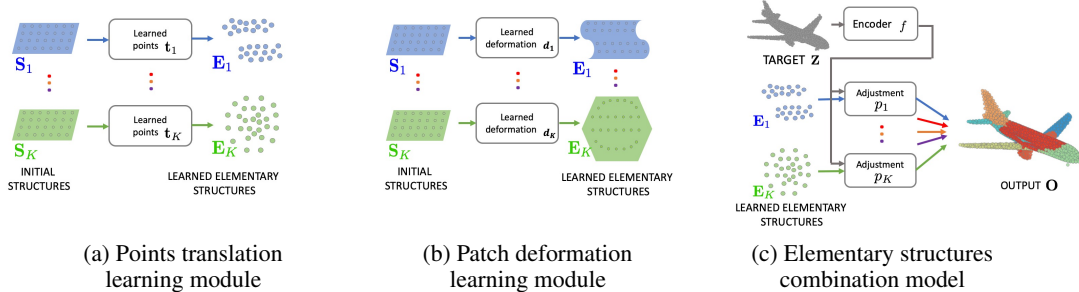

|  (a) Points translation learning module | (b) Patch deformation learning module | (c) Elementary structures combination model |

Figure 2: **Approach overview.** At training time, we learn (a) translations $t_i$ or (b) deformations $d_i$ that transform points from the unit square $S_i$ into shared learned elementary structures (c). At evaluation time, we transform each elementary structure $E_i$ to target shape $Z$ using learned shape-dependent adjustment networks $p_i$ that produce points on the surface of the output shape $O$.

across different shapes, and thus cannot be used to discover common shape structures or inter-shape relationships.

On the other side of the spectrum, instead of simple primitives, some techniques fit deformable mesh models [1, 2, 19, 32]. While they can capture complex structures, these techniques are also prone to being stuck in local optima, due to large number of degrees of freedom (e.g., mesh vertex coordinates).

Neural network architectures have been used to facilitate the mesh fitting [10], learning to predict the deformation of a template to reconstruct unstructured input point cloud. This approach is sensitive to the choice of the template. We demonstrate that our method improves the quality of the fitting by learning the structure of the reference shape. Neural mesh fitting has been also employed for geometrically and topologically diverse datasets that do not have a natural template. In these cases, meshed planes or spheres can be deformed into complex 3D structures [11, 30]. We extend this line of work by proposing a technique for learning the base shapes that are further used to approximate the shapes in the collection. Learning these elementary structures enables us to more accurately and consistently reconstruct the shapes in the collection.

## 3   Approach

We aim to learn shared elementary structures to reconstruct a set of 3D shapes. We visualize an overview of our approach in Figure 2. We formulate two ways to learn elementary structures – via *patch deformation learning* and *point translation learning* modules. The elementary structures are learned over the entire training set and do not depend on the input during testing. At test time, the elementary structures are deformed by *adjustment modules* to create the output 3D shape. These modules take as inputs features computed from the input via an encoder network and the coordinates of the elementary structure points and output the 3D coordinates of the deformed primitives.

For the task of 3D shape reconstruction, we assume that we are given a training set $\mathcal{Z}$ of target shapes $\mathbf{Z} \in \mathcal{Z}$. Our goal is to reconstruct the target shapes using a set of $K$ learned elementary structures $\mathbf{E}_1, \ldots, \mathbf{E}_K$, which are deformed via shape-dependent adjustment modules $p_1, \ldots, p_K$. We represent each shape by a feature vector $f(\mathbf{Z})$ computed by a point set encoder $f$ (defined later in this section). Each adjustment module $p_k$ takes as inputs the coordinates of a point in the associated elementary structure $\mathbf{e} \in \mathbf{E}_k$ and the feature vector of the target shape $f(\mathbf{Z})$ and outputs 3D coordinates of the corresponding point. The output shape $\mathbf{O} = p(\mathbf{Z})$ can thus be written as the union over learned and adjusted elementary structures,

$$\mathbf{O} = p(\mathbf{Z}) = \bigcup_{k=1}^{K} \bigcup_{\mathbf{e} \in \mathbf{E}_k} p_k(\mathbf{e}, f(\mathbf{Z})). \tag{1}$$

If the elementary structures were unit squares or a unit sphere, then this equation would describe exactly the AtlasNet [11] model. On the other hand, the 3D-CODED model [10] uses an instance of $\mathcal{Z}$ as a single elementary structure. Generalizing these approaches, our goal is to automatically learn

the elementary structures $\mathbf{E}_k$ over a shape collection. The intuition behind our approach is that if the elementary structures $\mathbf{E}_k$ have useful shapes to reconstruct the target, the adjustment $p_k$ should be easier to learn and more interpretable.

## 3.1 Learnable elementary structures

For each $k \in \{1, \ldots, K\}$, we start from an initial surface $\mathcal{S}_k$ on which we sample $N$ points to obtain an initial point cloud $\mathbf{S_k}$. We then pass each sampled point $\mathbf{s}_{k,i} \in \mathbf{S_k}$ for $i \in \{1, \ldots, N\}$ through elementary structure learning modules $\psi_k$.

We consider two types of elementary structure learning module $\psi_k$.

The first type, patch deformation learning module, learns a continuous mapping $d_k$ to obtain deformed points $\mathbf{e}_{k,i} = d_k(\mathbf{s}_{k,i})$ starting from sampled point $\mathbf{s}_{k,i}$. The intuition behind the deformation module is that elementary structures $E_k$ should be surface elements, and can thus be deduced from the transformation of the original surfaces $\mathbf{S}_k$. Alternatively, we consider a point translation learning module which translates independently each of the points $\mathbf{s}_{k,i}$ by a learned vector $\mathbf{t}_{k,i}$, $\mathbf{e}_{k,i} = \mathbf{t}_{k,i} + \mathbf{s}_{k,i}$. This module thus allows the network to update independently the position of each point on the surface. The result of either module results in a set of elementary structure points $\mathbf{e}_{k,i} = \psi_k(\mathbf{s}_{k,i})$, and we write the elementary structure $\mathbf{E}_k$ as the union of the independently deformed or translated points $\mathbf{s}_{k,i} \in \mathbf{S_k}$.

In Section 4 we will show that different choices here can be desirable depending on the application domain.

**Dimensionality of the elementary structures.** While it is natural to consider elementary structures as sets of 3D points, we can extend the idea to other dimensions. We experimented with 2D, 3D, and 10D elementary structures and show that while they are less interpretable, higher-dimensional structures lead to better shape reconstruction results.

## 3.2 Architecture details

The following describes more details of our final network.

**Shape encoder.** We represent the input shape as a point cloud, and we use as shape encoder a simplified version of the PointNet network [21] used in [10, 11]. We represent each 3D point of the input shape as a 1024 dimensional vector using a multi-layer perceptron with 3 hidden layers of 64, 128 and 1024 neurons and ReLU activations. We then apply max-pooling over all point features followed by a linear layer, producing a global shape feature used as input to the adjustment modules.

**Patch deformation learning module.** The patch deformation learning modules are continuous-space deformations that we learn as multi-layer perceptrons with 3 hidden layers of 128, 128 and 3 neurons and ReLU activations. This module takes as input coordinates of points in the initial structures and can compute not only a set of points [11] but the full image of a surface. If this module is used, we can densely sample points on the generated surface.

**Point translation learning module.** The point translation learning modules learn a translation for each of the $N$ points of the associated initial structure. While this step gives more flexibility than generating points through the patch deformation learning module, it can only be applied for a fixed number of points, similar to point-based shape generation [7].

**Adjustment module.** The goal of the adjustment modules $p_k$ is to reconstruct the input shape by positioning each elementary structure. The intuition is that this adjustment should be relatively simple. However, we can expect the quality of the reconstruction to increase using more complex adjustment modules. In this paper, we consider two cases:

- *Linear adjustment*: each adjustment module applies an affine transformation to the corresponding elementary structure. The parameters of this transformation are predicted by a multi-layer perceptron that takes as input the point cloud feature vector generated by the encoder. We use three hidden MLP layers (512, 512, 12), ReLU activation, BatchNorm layers and a hyperbolic tangent at the last layer for this module.

- *MLP adjustment*: each adjustment module uses a multi-layer perceptron (MLP) that takes as inputs the concatenation of the coordinates of a point from the associated elementary

| | Single-category training | | Multi-category training | | | | Multi-category training | | |
|---|---|---|---|---|---|---|---|---|---|
| | Airplanes | Chairs | Airplanes | Chairs | All | | | Points | Def. |
| *Linear adjustment* | | | | | | | *MLP adjustment* | | |
| AtlasNet [11] | 1.57 | 4.14 | 2.22 | 3.72 | 3.07 | | 2D | 1.28 | 1.42 |
| Deformation | 1.16 | 2.76 | 1.49 | 2.52 | 2.26 | | 3D | 1.22 | 1.43 |
| Points | 1.04 | 2.00 | 1.35 | 2.47 | 2.11 | | 10D | **1.21** | **1.39** |
| *MLP adjustment* | | | | | | | *Linear adjustment* | | |
| AtlasNet [11] | 0.91 | 1.64 | 0.81 | 1.50 | 1.45 | | 2D | 2.45 | 2.75 |
| Deformation | 0.87 | 1.56 | 0.81 | **1.25** | 1.43 | | 3D | 2.11 | 2.26 |
| Points | **0.79** | **1.43** | **0.71** | **1.25** | **1.22** | | 10D | **1.66** | **1.90** |

Table 1: **ShapeNet reconstruction.** We evaluate variants of our method for single- and multi-category reconstruction tasks. *Left*: Linear vs MLP adjustment, Patch Deformation vs Points Translation with 3D elementary structures. *Right*: different template dimensionality and deformation vs points learning modules in the multi-category setup with MLP-adjustement. We report Chamfer distance (multiplied by $10^{-3}$). AtlasNet uses 10 patch primitives, which is the same as our approach, without the learned elementary structures.

structure and the shape feature predicted by the shape encoder and outputs 3D coordinates. We use the same architecture as [11] for this network to obtain comparable results.

## 3.3 Losses and training

We now discuss two scenarios in which we tested our approach.

**Training with correspondences.** In this scenario, we assume point correspondences across all training examples and a common template that we can use as an initial structure for all shapes. More precisely, we assume that each training shape $\mathbf{Z}$ is represented as an ordered set of $N$ 3D points $\mathbf{z}_1, \ldots, \mathbf{z}_N$ in consistent locations on all shapes. Since all shapes are in correspondence, we consider a single elementary structure $S_1$ ($K = 1$) and N sampled points on the shape $\mathbf{s}_{1,1}, \ldots, \mathbf{s}_{1,N}$. We then train our network to minimize the following squared loss between sampled points $\mathbf{z}_i$ on each training shape to reconstructed points starting from sampled template points $\mathbf{s}_{1,i}$ :

$$\mathcal{L}_{\mathbf{sup}}(\theta) = \sum_{\mathbf{Z} \in \mathcal{Z}} \sum_{i=1}^{N} \|\mathbf{z}_i - p_1(\psi_1(\mathbf{s}_{1,i}), f(\mathbf{Z}))\|^2 \tag{2}$$

where $\theta$ are the parameters of the networks. Note that at inference, we do not need to know the correspondences of the points in the test shape, since they are processed by the point set encoder which is invariant to the order of the points. Instead, the points in the reconstruction shapes will be in correspondence with the elementary structure and by extension with each other. We use this property to predict correspondences between test shapes, following the pipeline of [10]. Learning the elementary structures is the difference between our approach and 3D-CODED [10] in this scenario, which leads to improved reconstruction and correspondence accuracy.

**Training without correspondences.** We are also able to train our system when no correspondence supervision is available during training. In this case, there are many options for our choice of elementary structures. To be comparable with AtlasNet [11], we will assume we have $K$ elementary structures and that each initial structure $\mathcal{S}_k$ is a unit 2D square patch. For a given training shape $\mathbf{Z}$, we compute the output shape $\mathbf{O} = p(\mathbf{Z})$ according to Equation 1, and train our network's parameters to minimize the symmetric Chamfer distance [7] between the point clouds $p(\mathbf{Z})$ and $\mathbf{Z}$.

$$\mathcal{L}_{\mathbf{unsup}}(\theta) = \sum_{\mathbf{Z} \in \mathcal{Z}} \sum_{\mathbf{z} \in \mathbf{Z}} \min_{k \in \{1,...,K\}, \, i \in \{1,...,N\}} \|\mathbf{z} - p_k(\psi_k(\mathbf{s}_{k,i}), f(\mathbf{Z}))\|^2$$
$$+ \sum_{\mathbf{Z} \in \mathcal{Z}} \sum_{k=1}^{K} \sum_{i=1}^{N} \min_{\mathbf{z} \in \mathbf{Z}} \|\mathbf{z} - p_k(\psi_k(\mathbf{s}_{k,i}), f(\mathbf{Z}))\|^2 \tag{3}$$

where $\theta$ are the parameters of the networks. In all of our experiments, we used $K = 10$.

**Training details.** We use the Adam optimizer with a learning rate of 0.001, a batch size of 16, and batch normalization layers. We train our method using input point clouds of 2500 points when correspondences are not available and 6800 points when correspondences are available. When training

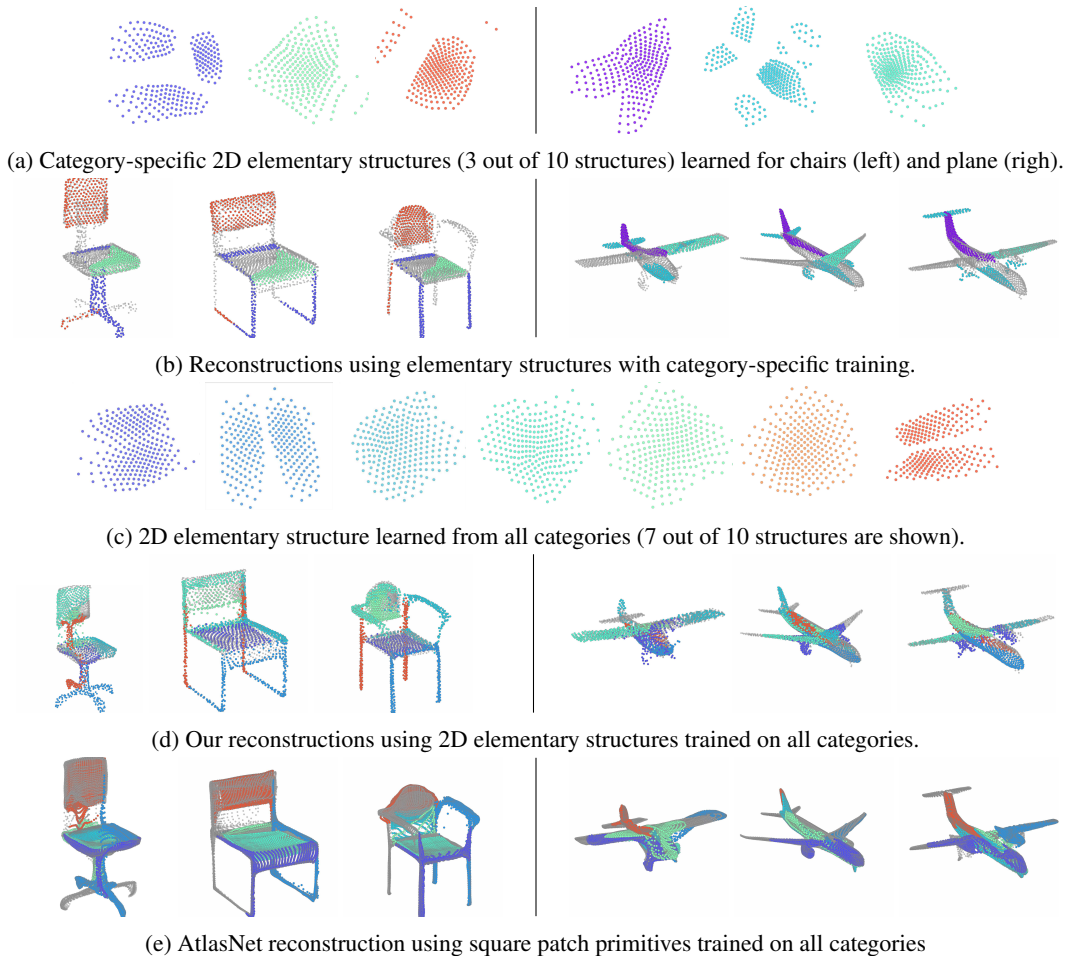

(a) Category-specific 2D elementary structures (3 out of 10 structures) learned for chairs (left) and plane (righ).

(b) Reconstructions using elementary structures with category-specific training.

(c) 2D elementary structure learned from all categories (7 out of 10 structures are shown).

(d) Our reconstructions using 2D elementary structures trained on all categories.

(e) AtlasNet reconstruction using square patch primitives trained on all categories

Figure 3: We visualize elementary structures using point learning and MLP adjustment modules. For all reconstruction results, we show in color the points corresponding to the visualized 2D primitives. For AtlasNet, the primitives are unit squares (so we do not show the elementary structures), and we visualize seven of them for the reconstruction (similarly to our method). Contrary to AtlasNet, our learned elementary structures have limited overlap in the reconstructions and better reconstructs the shapes.

using only the deformation modules $d_k$, we resample the initial surfaces $S_k$ at each training step to minimize over-fitting. At inference time, we sample a regular grid to allow easy mesh generation. We train our model on an NVIDIA 1080Ti GPU, with a 16 core Intel I7-7820X CPU (3.6GHz), 126GB RAM and SSD storage. Training takes about 48h for most experiments. Using the trained models from the official implementation on all categories, AtlasNet-25 performance is $1.56$ (see also Table 1 in the Atlasnet paper). Using the released code to train AtlasNet-10 yields an error of $1.55$. By adding a learning rate schedule to the original implementation we decreased this error to $1.45$ and report this improved baseline (see Table 1).

## 4   Experiments

In this section, we show qualitative and quantitative results of our approach on the tasks of shape reconstruction and shape matching.

### 4.1   Generic object shape reconstruction

We evaluate our approach on non-articulated generic 3D object shapes for the task of shape reconstruction. We use the training setting without correspondences described in Section 3.3.

**Dataset, evaluation criteria, baseline.** We evaluate on the ShapeNet Core dataset [6]. For single-category reconstruction, we evaluated over airplane (5424/1360 train/test shapes) and  chair

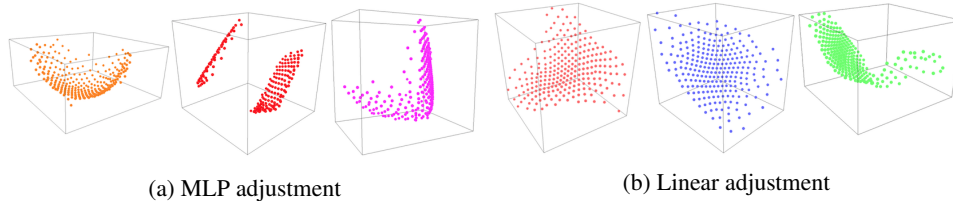

(a) MLP adjustment                 (b) Linear adjustment

Figure 4: Three (out of ten) learned 3D elementary structures learned by the point translation learning approach when training on all ShapeNet categories.

(3248/816) categories. For multi-category reconstruction, we used 13 categories –  airplane, bench, cabinet, car, chair, monitor, lamp, speaker, firearm, couch, table, cellphone, watercraft (31760/7952). We report the symmetric Chamfer distance between the reconstructed and target shapes. All reported Chamfer results are multiplied by $10^{-3}$. As a baseline, we compare against AtlasNet [11] with ten unit-square primitives.

**Single-category shape reconstruction.** For our first experiment, we trained separate networks for the different ShapeNet Core categories. Figure 3a demonstrates learned 2D elementary structures using ten 2D unit squares as initial structures $S_k$. In Figure  3b, we show shape reconstructions using our points translation learning module with MLP adjustments. Note the emergence of symmetric and topologically complex elementary structures.

**Multi-class shape reconstruction.** We now evaluate how well our method generalizes when trained on multiple categories, again using 2D elementary structures with point translation learning module and MLP-adjustements. As in single-category case, we observe discovery of non-trivial 2D elementary structures (Figure 3c) that are used to accurately reconstruct the shapes (Figure 3d), with higher fidelity than the baseline performance of AtlasNet with ten 2D square patches (Figure 3e). Note how AtlasNet is less faithful to the topology of reconstructed shapes, incorrectly synthesizing geometry in hollow areas between the back and the seat. Our quantitative evaluation in Table 1 confirms that AtlasNet provides less accurate reconstructions than our method.

**Linear vs MLP adjustment.** We evaluated networks trained in both the single- and multi-category settings with linear and MLP adjustment modules using 3D learned elementary structures (Table 1 left, Figure 4). In all experimental setups, we observe that the MLP adjustment offers significant quantitative improvements over restricting the network to use linear transformations of the elementary structures. This result is expected as linear adjustment allows only limited adaptation of the elementary structures for each shape. Similar to shape abstraction methods [28], linear adjustment allows a better intuition of the shape generation process but limits the reconstruction accuracy. Using MLP adjustments, however, offers the network more flexibility to faithfully reconstruct the shapes.

**Patch deformation vs points translation modules.**    We compare using patch deformation vs points translation modules in Table 1. The patch deformation learning module does not allow topological changes and discontinuities in mapping, and produces inferior results in comparison to points translation learning. On the other hand, learning patch deformations enables the estimation of the entire deformation field. Thus one can warp an arbitrary number of points or even tessellate the domain and warp the entire mesh to generate the polygonal surface, which is more amenable to tasks such as rendering.

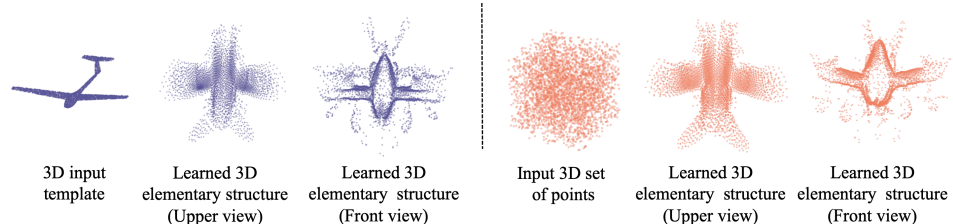

| 3D input template | Learned 3D elementary structure (Upper view) | Learned 3D elementary  structure (Front view) | Input 3D set of points | Learned 3D elementary  structure (Upper view) | Learned 3D elementary  structure (Front view) |

Figure 5: 3D elementary structure obtained with point learning when initializing the training from a template shape (left) or a random set of points (right). See text for details.

**Higher-dimensional structures.** We experimented with the dimensionality of the learned elementary structures. Figures 3a and 3c suggest that learned 2D elementary structures can capture interesting topological and symmetric aspects of the data – splitting, for instance, the patch into two identical parts for the legs of the chairs. note also the variable point density. Similarly, learned 3D elementary structures with linear adjustment and patch deformation learning modules are shown in Figure 1 for the airplane category. Note that they roughly correspond to meaningful parts, such as wings, tail and reactor. Figure 4 shows 3D elementary structures inferred from all ShapeNet categories, where the learned structures include non-trivial elements such as symmetric planes, sharp angles, and smooth parabolic surfaces. The learned structures are often correspond to consistent parts in the reconstructions. In our quantitative evaluations (Table 1, right) we found that the results improve with the dimensionality. The improvement diminishes for higher-dimensional spaces and are more difficult to visualize and interpret.

**Consistency in template elementary structures.** We experimented with several initializations of our elementary structures on the ShapeNet plane category. We used the point translation learning method and a single 3D elementary structure. In Figure 5, we show our results when initializing the elementary structure with either a plane 3D model (left) or a set of random 3D points sampled uniformly (right). Notice that the learned 3D elementary structure is similar regardless of the initial template shape.

**Generalization to new categories.** To test the generality of our approach, we trained on the chair category using ten 2D elementary structures and tested on the table category. As shown in Figure 6, point translation learning outperforms both patch deformation learning and AtlasNet. Figure 7 shows qualitatively how the elementary structures are positioned on chairs and tables. Notice how the chair and table legs are reconstructed by the same elementary structures.

|  | Chairs | Table |
|---|---|---|
| AtlasNet | 1.64 | 4.70 |
| Patch. | 1.56 | 4.82 |
| Point. | **1.34** | **4.45** |

Figure 6: **Category generalization.** Chamfer distance for networks trained on chairs and tested on either the chairs or tables test sets.

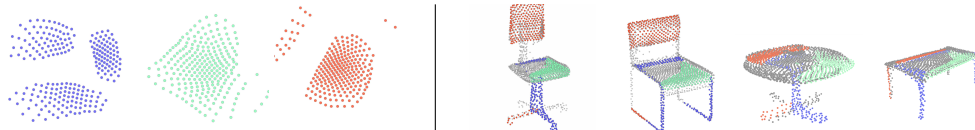

Figure 7: Elementary structures learned on chairs (**left**) used to reconstruct chairs and tables (**right**).

**Number of parameters.** In Figure 8, we show the number of parameters for AtlasNet and our method. Our method has less than 1% additional parameters to learn the elementary structures – $2.0 \times 10^6$ and $2.5 \times 10^3$ for patch deformation and point translation, respectively (orders of magnitude smaller than $1.8 \times 10^8$ for the full network). During inference, our approach has the same complexity as AtlasNet as the elementary structures are precomputed and remain fixed for all shapes. We also tried training AtlasNet with six layers (6-layer AN), which significantly increases the number of parameters. Our approach with points translation learning outperforms all methods.

|  | Param. | Chamfer |
|---|---|---|
| AtlasNet | $1.8 \times 10^8$ | 1.45 |
| 6-layer AN | $3.9 \times 10^8$ | 1.35 |
| Patch. | $1.8 \times 10^8$ | 1.43 |
| Point. | $1.8 \times 10^8$ | **1.22** |

Figure 8: Impact of number of parameters on reconstruction error.

### 4.2 Human shape reconstruction and matching

We now evaluate our approach on 3D human shapes for the tasks of shape reconstruction and matching using the training setup with correspondences described in Section 3.3. For this task, we use a single elementary structure for the human body using one of the meshes as the initial structure $\mathbf{S}_1$. Since we use a single elementary structure and the shapes are deformable, we only report results using the MLP-adjustment.

**Datasets, evaluation criteria, baselines.** We train our method using the SURREAL dataset [29], extended to include some additional bend-over poses as in 3D-CODED [10]. We use 229,984 SURREAL meshes of humans in various poses for training and 224 SURREAL meshes to test reconstruction quality. To evaluate correspondences on real data, we use the FAUST benchmark [5] consisting of 200 testing scans with $\sim 170k$ vertices from the "inter" challenge, including noise and holes which are not present in our training data. As a baseline, we compared against 3D-CODED [10].

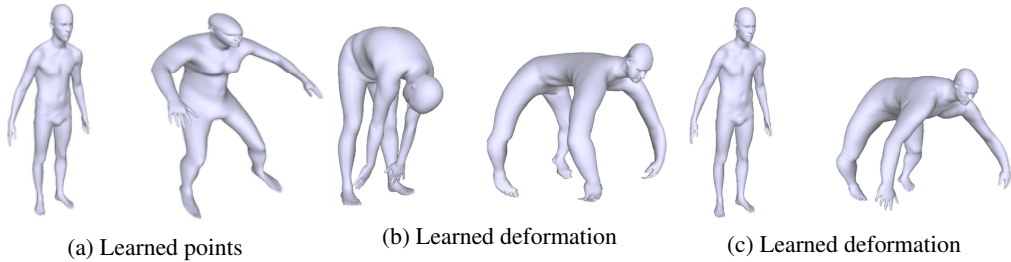

(a) Learned points          (b) Learned deformation          (c) Learned deformation

Figure 9: Initial shape (left) and learned elementary structure (right) using the deformation or points learning modules. Notice the similarity between the elementary structure learned with the different approaches.

| | SURREAL [29] | FAUST [5] Inter \| Intra | | SURREAL [29] | |
| | | | | Points | Deform. |
| --- | --- | --- | --- | --- | --- |
| 3D-CODED | 1.32 | 2.64 \| 1.747 | 2D | 1.54 | 6.76 |
| Deformation | 1.44 | 2.58 \| 1.742 | 3D | 1.00 | 1.44 |
| Points | 1.00 | 2.71 \| 1.626 | 10D | 1.06 | 1.18 |

Table 2: **Human correspondences and reconstruction.** We evaluate different variants of our method (with deformation vs points translation learning and different template dimensionality) for surface reconstruction (SURREAL column) and matching (FAUST column). We report Chamfer loss for the former and correspondence error for the latter (measured by the distance between corresponding points). Results in the left table are with 3D elementary structures, and the only difference with the 3D-CODED baseline is thus the template/elementary structure learning. The table on the right shows results with elementary structures of different dimensions.

**Results.** Figure 9 shows learned elementary structures using deformation or points translation learning and different initial surfaces. We observe that the learned templates are inflated, bent, and with their arm and legs in a similar pose, suggesting a reasonable amount of consistency in the properties of a desirable primitive shape for this task.

As before, we found that points translation learning provides the best reconstruction (see SURREAL column in Table 2). Both of our approaches also provide lower reconstruction loss than 3D-CODED.

We used reconstruction to estimate correspondences by finding closest points on the deformed elementary structure as in 3D-CODED [10]. We report correspondence error in the "FAUST" column in Table 2. We observe that deformation learning provides better correspondences than points learning, also yielding state-of-the-art results and clear improvement over 3D-CODED. This result is not surprising because understanding the deformation field for the entire surface is more relevant for matching and correspondence problems.

**Elementary structure dimension.** Similar to generic object reconstruction, we evaluate with 2D, 3D and 10D elementary structures (Table 2, right). Note that when using the patch deformation learning module we control the output size and therefore it is easy to map the input 3D template to higher- or lower-dimensional elementary structure. On the other hand the points translation learning module does not allow to change dimensionality of the input template. Hence, for 2D elementary structures we project the 3D template (front-facing human in a T-pose) to a front plane, and for 10D elementary structures we embed the 3D human into a hyper-cube, keeping higher dimensions as zero. The difference in performance is clearer for human reconstruction than for generic object reconstruction, which can be related both to the fact that humans are complex with articulations and that we use a single elementary structure for all human reconstructions.

## 5   Conclusion

We have presented a method to take a collection of training shapes and learned common elementary structures that can be deformed and composed to consistently reconstruct arbitrary shapes. We learn consistent structures without explicit point supervision between shapes and we demonstrate that using our structures for reconstruction and correspondence tasks results in significant quantitative improvements. When trained on shape categories, these structures are often interpretable. Moreover, our deformation learning approach learns elementary structures as the deformation of continuous surfaces, resulting in output surfaces that can densely sampled and meshed at test time. Our approach opens up possibilities for other applications, such as shape morphing and scan completion.

**Acknowledgments.** This work was partly supported by ANR project EnHerit ANR-17-CE23-0008, Labex Bézout, and gifts from Adobe to École des Ponts.

## Footnotes

[1]http://imagine.enpc.fr/ deprellt/atlasnet2

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
