[Reviews · NeurIPS 2019]

Reviewer 1



Summary: This paper states an interesting and novel idea- that learned shape bases could outperform hand-crafted heuristic functions. On the flip side, though, the method and experimental setups make drawing clear conclusions difficult, diminishing the impact. Post-rebuttal: The authors gave convincing responses to questions about the atlasnet comparison and about the number of parameters. So the final review is increased from 6 to 7. Originality: -As stated above, the paper has an interesting and novel high-level key idea. Whether the proposed method is really learning shape bases rather than using heuristic bases is a matter of interpretation, though. The input to the method is still K unit squares or a human template. So whether the output bases are really learned or are just fixed intermediate activations is a subtle difference, especially given that they can be >3D and are treated only as features in the MLP case. Particularly in the multiclass case the elementary structures don’t look very much like their output deformations. Is there a more convincing demonstration that the difference is more than just extra degrees of freedom in the model? -There is a missing important related work: "Learning Category-Specific Mesh Reconstruction from Image Collections" by Kanazawa et al. This work learns a mean shape per class that is a deformation of a sphere to do reconstruction, which is quite related to learning consistent deformations of squares per structured component to do reconstruction. Quality: It's currently unclear whether the AtlasNet result should be treated as a true ablation or as a baseline comparison. On one hand, AtlasNet10 seems quite similar to the method, but without the extra learning module layers. On the other hand, it isn't exactly the same, and the paper calls it a baseline. It seems really important to show both results. First, two real ablations of the method. One that just removes the learning modules, and another that replaces them with extra parameters to prove that the improvement in performance is not just coming from extra degrees of freedom. Second, a real comparison to AtlasNet- the authors provide pretrained weights for 25 squares on point cloud inputs. Comparing to this would be a much more trustworthy baseline establishing a clear performance improvement. The claim that the elements correspond to parts or an abstraction seems questionable. The elements are not internally connected in point learning; for example the teal airplane structure learned in figure 3a) contains engines, fuselage, and the tail, while the purple part is more tail, more fuselage, and part of the wing. In figure 4d) the teal part is all wing in one example but describes half the fuselage in another. What do the reconstructions and decompositions look like for examples with greater structural variation? I.e. fighter jets, biplanes, etc for planes. The figure 5) result is quite interesting- the convergent behavior regardless of initialization indicates there’s some kind of useful warp to the initial template. Would this consistency hold for the ShapeNet case? Clarity: Is the evaluation in Table 1 on the train set or test set? This review assumes test, but it isn’t very clear since it says ’single-category training’ and ‘multi-category training’. In the case it is the training set, please explain why the performance differences could not just be explained by the parameter count of each model. Significance: The paper could definitely be significant to future research if it were clear that there is improved performance which is attributable to learning shape building blocks. Currently the results are promising, but not conclusive enough to establish a real win. Also the method still needs heuristic bases, such as an input human mesh or K squares, which diminishes the significance of the proposed 'learned' elementary structures somewhat.

Reviewer 2



The key idea of this paper introduces a new deformable part model for 3D shape reconstruction. The key idea is to make the deformable model learnable. The approach is not brand-new, but it manages to put all the pieces together. The paper also has a very good evaluation session, with very detailed components on baseline approaches and evaluation metrics, satisfactory results, and some analysis. On the down side, the paper can be made stronger, by analyzing the proposed approach deeper. For example, it is still that clear why the proposed neural network model can learn good deformation model? It can benefit from a more detailed analysis.

Reviewer 3



Summary This paper proposes a pipeline, decomposing/modeling 3D shapes by learned elementary structures, which can be used in shape generation and matching. The authors build their pipeline based on AtlasNet, with elementary structure learning modules so that all the elementary structures are not fixed but learned from data. Two approaches are then introduced, one based on deformation, and another directly learns the translation for each of the points. The authors then discuss the design of loss function: if we have point correspondences across training examples, then we can use the trivial squared error; if not, we can use Chamfer distance as the loss. Finally, the authors demonstrate the performance of the proposed model, by doing shape reconstruction on ShapeNet dataset and doing human shape reconstruction and matching on SURREAL dataset. Strengths -The idea of learning elementary structures from data is novel. By letting the model learn from data, with higher probability, the model will be able to learn some meaningful structures. -The results look impressive. As shown in figure 3, the proposed method successfully learned some meaning structures (e.g., tailplane in figure 3(b)). Weaknesses -Need to improve the readability. For example, the notations and names of modules are kind of confusing. In figure 2, t_1 to t_K are bold while d_1 to d_K are not bold. In line 101, p_1 to p_K are called positioning modules while in line 144 they are called adjustment modules. Making all of them consistent would help readers to understand the paper more easily. Comments after the rebuttal ************************************ Thank the authors for the rebuttal. The results in Figure 2 looks good, but still not particularly amazing. So I kept my rating.

[Author Response · NeurIPS 2019]

# Learning Elementary Structures for 3D shape generation and matching

We thank the reviewers (Rs) for their comments. We are pleased to receive the positive reviews. If accepted, we will incorporate all feedback in the final version.

**Generalization to new categories. (R1, R2, R3)**   To test the generality of our approach, we followed the reviewers' suggestion and trained on chairs using 10 2D elementary structures and tested on tables. As shown in Figure 1 (this rebuttal), point learning outperforms both transformation learning and AtlasNet trained with 10 patches - all Chamfer results in the rebuttal are multiplied by $10^{-3}$. Figure 2 (this rebuttal) shows qualitatively how the elementary structures are positioned on chairs and tables. Notice how the chair and table legs are reconstructed by the same elementary structures.

|  | Chairs | Table |
|---|---|---|
| AtlasNet | 1.64 | 4.70 |
| Transfo. | 1.56 | 4.82 |
| Point. | **1.34** | **4.45** |

Figure 1: **Generalization.** Chamfer loss results of the networks trained on chairs and tested on either the chairs or tables test set.

Figure 2: Elementary structures learned on chairs (**left**) used to reconstruct chairs and tables (**right**).

**Where does the performance boost come from? (R1-II, R2)**   In Figure 3 (this rebuttal), we show the number of parameters for AtlasNet and our method. Our method has less than 1% additional parameters to learn the elementary structures – $2.0 \times 10^6$ and $2.5 \times 10^3$ for transformation and point learning, respectively (notice that they are orders of magnitude smaller than $1.8 \times 10^8$). During inference, our approach has the same complexity as AtlasNet as the elementary structures are precomputed and remain fixed for all shapes. As suggested, we also tried training AtlasNet with 6 layers (6-layer AN), which significantly increases the number of parameters. Our approach with points learning outperforms all methods.

|  | Param. | Chamfer |
|---|---|---|
| AtlasNet | $1.8 \times 10^8$ | 1.45 |
| 6-layer AN | $3.9 \times 10^8$ | 1.35 |
| Transfo. | $1.8 \times 10^8$ | 1.43 |
| Point. | $1.8 \times 10^8$ | **1.22** |

Figure 3: **Impact of number of parameters on reconstruction error.**

**Consistency in template elementary structures. (R1-I)**   We extended the experiment with SURREAL from Figure 5 of our paper to the plane category of ShapeNet using the point learning method. We used a single 3D elementary structure as in the SURREAL experiment. In Figure 4 (this rebuttal), we initialized the elementary structure with either a plane 3D model (**left**) or a set of random 3D points sample uniformly (**right**). Notice that (1) the learned 3D elementary structures are consistent regardless the template shape and (2) we do not need to input heuristic basis functions since using a set of random 3D points give similar results.

| 3D input template | Learned 3D elementary structure (Upper view) | Learned 3D elementary structure (Front view) | Input 3D set of points | Learned 3D elementary structure (Upper view) | Learned 3D elementary structure (Front view) |

Figure 4: **Robustness of the learned 3D elementary structure.**

**Comparison to AltasNet-trained models. (R1-III)**   Using the trained models from the official implementation on all categories, AtlasNet-25 performance is 1.56 (see also Table 1 in the Atlasnet paper). Using the released code to train AtlasNet-10 yields 1.55 of performance. In our paper, we added a learning rate schedule to the original implementation and got an error of 1.45 (see Table 1 of our paper). Using the same learning rate schedule, PointLearning-10 and TransformationLearning-10 perform, respectively, 1.22 and 1.43. For reference, PointLearning-25 and TransformationLearning-25 perform, respectively, 1.21 and 1.40 – a significant 22% and 9% boost.

**Details, References, Writing. (R1, R2, R3)**   Results in Table 1 (from the paper) are evaluated on the test set (**R1**). We will mitigate the claim that elementary structures correspond to semantic parts (**R1**), add missing discussion on Kanazawa et al. (**R1**) and improve the consistency of the notations (**R3**).

[Meta-Review · NeurIPS 2019]

All reviewers recommended accept.